# Diversity and Antimicrobial Activity of Intestinal Fungi from Three Species of Coral Reef Fish

**DOI:** 10.3390/jof9060613

**Published:** 2023-05-26

**Authors:** Xinyu Liao, Jiadenghui Yang, Zanhu Zhou, Jinying Wu, Dunming Xu, Qiaoting Yang, Saiyi Zhong, Xiaoyong Zhang

**Affiliations:** 1University Joint Laboratory of Guangdong Province, Hong Kong and Macao Region on Marine Bioresource Conservation and Exploitation, College of Marine Sciences, South China Agricultural University, Guangzhou 510642, China; liaoxinyu2023@163.com (X.L.); 19120536855@163.com (J.Y.); wujinyingscau@126.com (J.W.);; 2Guangdong Provincial Key Laboratory of Aquatic Product Processing and Safety, College of Food Science and Technology, Guangdong Ocean University, Zhanjiang 524088, China; 3Technical Center of Xiamen Customs, Xiamen 361026, China; zzh_523@163.com (Z.Z.); dunmingxu@163.com (D.X.)

**Keywords:** antimicrobial activity, coral reef fish, diversity, intestinal fungi

## Abstract

Although intestinal microbiota play crucial roles in fish digestion and health, little is known about intestinal fungi in fish. This study investigated the intestinal fungal diversity of three coral reef fish (*Lates calcarifer*, *Trachinotus blochii*, and *Lutjanus argentimaculatus*) from the South China Sea using a culturable method. A total of 387 isolates were recovered and identified by sequencing their internal transcribed spacer sequences, belonging to 29 known fungal species. The similarity of fungal communities in the intestines of the three fish verified that the fungal colonization might be influenced by their surrounding environments. Furthermore, the fungal communities in different intestines of some fish were significantly different, and the number of yeasts in the hindgut was less than that in fore- and mid-intestines, suggesting that the distribution of fungi in fishes’ intestines may be related to the physiological functions of various intestinal segments. In addition, 51.4% of tested fungal isolates exhibited antimicrobial activity against at least one marine pathogenic microorganism. Notably, isolate *Aureobasidium pullulans* SCAU243 exhibited strong antifungal activity against *Aspergillus versicolor*, and isolate *Schizophyllum commune* SCAU255 displayed extensive antimicrobial activity against four marine pathogenic microorganisms. This study contributed to our understanding of intestinal fungi in coral reef fish and further increased the library of fungi available for natural bioactive product screening.

## 1. Introduction

Coral reef ecosystems are famous for their great biodiversity and ecological and economic value [1]. Although representing only 0.2% of the world’s ocean area proportionally, they provide habitats for at least 25% of known marine species [2,3]. In recent decades, a series of natural and anthropogenic disturbances have led to severe degradation of coral reefs [4]. As one of the most important communities in coral reef ecosystems, coral reef fish significantly contribute to ecosystem functioning and reef resilience [5,6]. At the same time, the microbiome, the collection of microorganisms associated with animals, is essential for the optimal growth and survival of the host fish [7,8]. Specifically, it plays a crucial role in host digestion, energy, and health [9,10]. Therefore, the microbial flora residing in the fish intestine has been studied in many species of coral reef fish, which greatly enriches our understanding of the complex interactions that occur between microorganisms residing in the fish intestine and their hosts [11].

Recently, the intestinal bacteria of coral reef fish have been researched intensively, which has revealed variable core flora among the different fish species. For example, in Golden Pompano (*Trachinotus ovatus*), Proteobacteria, Tenericutes, Spirochaetes, and Firmicutes were the dominant phyla [12]. In Barramundi (*Lates calcarifer*), Firmicutes was the most abundant and diverse phylum, while at the genus level, *Acinetobacter*, *Citrobacter*, *Exiguobacterium*, and *Pseudomonas* were the predominant genera [13]. In Mangrove Red Snapper (*Lutjanus argentimaculatus*), Proteobacteria and Firmicutes were the predominant phyla, and *Vibrio*, *Morganella*, *Schewanella*, *Photobacterium*, *Pantoea,* and *Bacillus* were the major genera [14]. Although the bacterial species in the intestines of different fish are not identical, the phylum Proteus, Firmicutes, and Bacteroides have been found to account for 90% of the studied intestinal microbial communities of fish, indicating that these bacteria may have an important influence and role in the intestinal function of fish. Further, microorganisms in the fish intestine can be roughly divided into two parts: autochthonous (being able to colonize in the gut) and allochthonous (considered to be free-living) [15]. From the host’s point of view, autochthonous bacteria are considered to be more important than allochthonous, as they promote the digestion and absorption of nutrients, stimulate the maturation and function of the immune system, and resist the invasion and infection of pathogens in three aspects [16]. These studies have contributed to our understanding of the species and composition of bacteria in the intestines of different coral reef fish. A recent study reported that the environmental microbiota that live in complex and dynamic seawater might affect the microorganisms in the intestine of fish [17]. Although there is still no definitive explanation for how microbial component taxa aggregate from their regional species pools, a host of possible drivers have been hypothesized, involving physiological host effect habitat filters, stochastic processes of colonization and migration, and different microbial competition processes [18].

As a diversified and widespread group of eukaryotes, fungi are major decomposers of microorganisms [19]. They should be regarded as essential parts of the intestinal microbiota due to their ability to maintain the normal functioning of physiological processes and tissue health and promote the proper progression of the immune system [20,21]. A recent study revealed that the fungal communities were not only related to fish species specificity but also to the respective physiological functions of different intestinal segments [19]. Among fungi, yeasts have been recognized as part of the normal microbiota of fish. Genus *Rhodotorula* was relatively frequent in both freshwater and marine fish, and the genus *Debaryomyces* were dominant in rainbow trout [22]. Caruffo et al. reported that many yeasts isolated from the gut of fish could be potential probiotics, reducing the mortality associated with the *Vibrio anguillarum* challenge [23]. However, the role of yeasts in fish health and nutrition is still not very clear [24]. Although many new methods of molecular biology, such as DNA microarray technology [25], fluorescence in situ hybridization [26], and high-throughput sequencing [27], can comprehensively analyze the species and an abundance of information of intestinal fungi, acquiring pure cultures of fungi is a critical step in the study of microbial function.

Barramundi (*L. calcarifer*), Golden Permit (*T. blochii*), and Mangrove Red Snapper (*L. argentimaculatus*) are three common and carnivorous coral reef fish in the South China Sea. These coral reef fish generally have a complex life history. Spawning and larval settlement occurred in the marine areas of river mouths, and then juvenile fish migrated upstream to fresh water, where they grew and became mature as males [28]. They exhibited a relatively high trophic status in the food chain and played a crucial role in maintaining the ecological balance of coral reef systems. They fed on fish and crustaceans but consumed a variety of algae as well [2].

In this research, we surveyed and compared the diversity and antimicrobial activity of culturable fungi associated with different intestinal segments of the three coral reef fish and their living seawater. Based on the internal transcribed spacer (ITS) sequences of culturable fungi that are associated with the fish intestine and seawater, we analyzed the phylogenetic diversity of these fungi. Furthermore, we also calculated the species and quantity in different intestinal segments separately and compared the dissimilarity of the fungal communities in the fish intestines and their living seawater. The information on fungal diversity and distribution in different intestinal segments of coral reef fish is very scarce and important, which provides the baseline data for gut microbiota in coral reef fish from the South China Sea and a reference for the study on gut microbiota in other marine organisms. In addition, 37 representative fungal isolates were selected for screening their antimicrobial activity against six marine pathogenic microorganisms, which can check the potential of these fungi from different intestinal segments in the production of bioactive molecules against pathogenic microorganisms and provide a good resource for our subsequent screening of marine microbial active substances.

## 2. Materials and Methods

### 2.1. Sample Collection and Preparation

Three species of coral reef fish and their surrounding environments (seawater) were collected from Daya Bay (114°32″59 E, 22°40″37 N), Shenzhen, South China, in August 2020. Located in the northeast of the South China Sea, the bay is a closed bay of 650 square kilometers with an annual average temperature of 22 degrees Celsius. The maximum sea surface temperature (SST) is 30 °C, and the minimum temperature is 15 °C [29]. 

All fish were mature, and males (with a length of 20~30 cm), and obtained by sea fishing and identified by Dr. Xiao Chen (South China Agricultural University, Guangzhou, China) as belonging to *L. argentimaculatus* (Samples LA1~LA3), *T. blochii* (Samples TB1~TB3), and *L. calcarifer* (Samples LA1~LA3), and were categorized as coral reef fish according to their habitat in reference to descriptions from Fish Base (https://www.fishbase.in/search.php (accessed on 18 March 2019), Copenhagen, Denmark) [2]. Three replicates of each fish species were sampled, and a total of nine samples from the three fish species was collected (Samples LA1~LA3, TB1~TB3, and LA1~LA3). Moreover, the phylogenetic tree was constructed to show the evolutionary relationships between the three species of fish based on the sequences of the cytochrome c oxidase subunit I (COI) genes downloaded from the National Center for Biotechnology Information (NCBI, Bethesda, MD, USA). The results demonstrated that *L. argentimaculatus* is less closely related to *T. blochii* and *L. calcarifer* (Figure 1). Three replicate seawater samples around the three fish were collected below the water surface using sterile seawater samplers (wiped with 75% alcohol before use). The depth and temperature of the three sets of seawater sampling sites were 5–10 m and 26 ± 1 °C, respectively [30].

Obtained fish were immediately put into sterile Ziploc plastic bags onboard, placed in a cool box during transportation to the laboratory, and kept at 4 °C before isolation (time from collection to isolation <12 h). Fish intestinal samples and seawater samples were collected and transported to the laboratory as soon as possible. 

### 2.2. Fungal Isolation of Fish Intestines and Seawater

The obtained fish were placed in a dissecting tray, and the body surface of the fish was wiped with 75% alcohol. After each fish was cut in an arc along the anus in an upward direction with dissecting scissors, its intestine was separated and divided into fore-, mid- and hind-intestinal segments with dissecting scissors [31]. Then the mucosa (including intestinal contents) of each intestinal segment was individually collected with sterile dissecting scissors, and finally, 0.5 g of the mucosa was gently scraped into a centrifuge tube [32].

Then, 0.5 mL of each of the three seawater samples was diluted into 4.5 mL of sterile seawater. For each of the 27 fish intestinal samples, 1 mL of sterile water and 0.1 g of intestinal sample were added to a mortar, then an appropriate amount of sterilized quartz sand was added and ground well with a grinding rod (the sand serves to make the sample fully ground), and 0.1 mL of the mixture was taken into the sterile seawater (0.9 mL) and mixed well [2].

Treated seawater samples and fish intestinal samples were diluted with sterile seawater at the ratio of 1:10 and 1:100. Then 100 µL aliquots were plated on six different isolation media (including GPA (1% Glucose, 0.1% peptone, 0.1% K_2_HPO_4_, 0.025% MgSO_4_, 2% agar), GYMA (0.4% Glucose, 0.4% yeast extract, 0.5% malt extract, 2% agar), GPSA (1% Glucose, 0.1% peptone, 1% starch, 0.1% K_2_HPO_4_, 0.1% MgSO_4_, 2% agar), GYPA (0.5% Glucose, 0.1% yeast extract, 0.5% peptone, 2% agar), PDA (20% Potato, 2% glucose, 2% agar), and SYA (1% Starch, 0.5% yeast extract, 2% agar)) with a sterile (sterilized by autoclaving at 121 °C for 30 min) glass spreading rod, and cultured in a constant temperature incubator at 26 °C until the morphology of the isolates could be observed clearly (The approximate days took to see the morphology of isolates was 5–7 days) [30]. Censored isolates were transferred to new PDA (200 g of diced potato, 20 g of glucose, 20 g of agar, and 1 L of seawater) medium individually, and the appearance and quantity of the isolated fungi were recorded and purified gradually until single colonies were obtained.

### 2.3. Extraction of Fungal Genomic DNA

The extraction of fungal genomic DNA was performed according to a published procedure [33]. Briefly, about 0.5–1.0 g of fresh mycelium of each selected fungal isolate and liquid nitrogen were taken into a pre-cooled mortar and ground thoroughly. The ground mycelium was transferred to a centrifuge tube, and 600 μL of sodium dodecyl sulfate (SDS) buffer was added to it, and the mixture was put in a water bath at 65 °C for 45 min and then centrifuged at 12,000 r/min (revolutions per minute) for 5 min. The supernatant was pipetted and transferred to a new centrifuge tube, and an equal volume of saturated phenol.,: chloroform: isoamyl alcohol with a volume ratio of 25:24:1 was added to the centrifuge tube. The mixture was slowly shaken and centrifuged at 12,000 r/min for 5 min. Then the supernatant was taken to a new centrifuge tube, and an equal volume of isopropanol was added to the centrifuge tube. The centrifuge tube was precipitated at −20 °C for 30 min and centrifuged at 12,000 r/min for 20 min. After that, the supernatant was removed; the DNA pellet was rinsed with 70% alcohol and inverted in the ultra-clean table to make the alcohol evaporate. Then the 20 μL TE buffer (10 mmol/L Tris-HCl, 1 mmol/L EDTA, pH 8.0) was added to the tube to dissolve the DNA, and the integrity of the remaining DNA was detected by electrophoresis and stored at −20 °C.

### 2.4. PCR Amplification of Target DNA and Sequencing of ITS Fragments

PCR amplification was performed using primers ITS1(5’-TCCGTAGGTGAACCTGCGG-3’) and ITS4 (5’-TCCTCCGCTTATTGATATGC-3’) for the extracted fungal genome [34]. ITS-r DNA amplification was amplified as a 25 μL PCR reaction system: 0.25 (10 μM) μL each for ITS1 and ITS4, 1 μL template DNA, 0.75 μL DMSO, 12.5 μL Taq premix (TakaRa, Beijing, China) and 10.25 μL H_2_O. Reaction conditions: after pre-denaturation at 95 °C for 5 min, amplification was performed with 30 cycles, denaturation at 94 °C for 45 s, annealing at 50 °C for 45 s, extension at 72 °C for 45 s, and extension at 72 °C for 10 min. PCR amplification products were detected by 1.2% agarose gel electrophoresis. The PCR amplification products were sequenced (Sanger sequencing method) on an ABI 3130 Genetic Analyzer (Applied Biosystems, Foster City, CA, USA) using the ITS1-ITS4 primers and BigDye Terminator V3.1 cycle sequencing kit (Applied Biosystems) [35].

### 2.5. Sequence Alignment and Phylogenetic Analyses

All sequencing results were analyzed for shearing using Mega 6.0 software, and the sequences were matched with those in the GenBank database using BLAST (a sequence alignment engine) at NCBI. The species name was assigned to the selected isolate when the top three matching BLAST hits were from the same species and were ≥95% similar to the query sequence. All ITS sequences were aligned using Clustal W in MEGA 6.0. Phylogenetic analysis of ITS sequences was carried out using the neighbor-joining method with 1000 bootstrap iterations of MEGA 6.0 software [36].

### 2.6. Bioassay of Antimicrobial Activity of Fungal Isolates

Six species of marine pathogenic bacteria and fungi were selected to test the antimicrobial activity of fungal isolates obtained from the intestines of three coral fish and their surrounding environments. Three species of marine pathogenic bacteria included *Vibrio alginolyticus* (UST981130-062, VA), *Pseudoaltermonas piscida* (UST010723-006, PP), and *Micrococcus luteus* (UST950701-006, ML) [37]; and three species of marine pathogenic fungi included *Aspergillus versicolor* (SCSGAF0096, AV), *A. sydowii* (SCSGAF0035, AS), and *Penicillium citrinum* (SCSGAF0192, PC) [38]. 

Selected and tested fungal isolates were grown on PDA plates and incubated for seven days at 26 °C. Agar blocks containing the strains were cut into discs and placed on analysis plates lined with pathogenic microorganisms. Pathogenic bacteria were cultured on LBA (0.5% peptone, 0.3% yeast extract, 2% agar) plates for 18 h at 30 °C, and pathogenic fungi were cultured on PDA (20% potato, 2% glucose, 2% agar) plates for 72 h at 26 °C. The antimicrobial activity of the experimental strains was represented by measuring the diameter of the growth inhibition zone (mm), and each antimicrobial assay was conducted in triplicate.

### 2.7. Data Analysis

In order to analyze the differences in fungal communities from different intestinal segments of multiple fish species and their surrounding environments, Bray–Curtis dissimilarity was chosen in this study. The Bray–Curtis analysis can show a linear response to the transfer of abundance from a given species in one plot to the same species in another plot in which the species is less abundant. In addition, five other coefficients can show a rather gradual, although nonlinear, change along with the transfer of abundances [39]. Recently, this method has been widely applied in the study of the analysis of differences in microbial communities [31,40]. The species Bray–Curtis coefficient was calculated from the presence (represented by 1)/absence (represented by 0) matrix of the fungi separated from the three fish intestines and the corresponding seawater, using SPSS software for Windows (Version 11.5) [31,40].

### 2.8. Nucleotide Sequence Accession Number

The ITS sequences of the 29 representative fungal isolates obtained in this experiment were registered in GenBank. Accession numbers are OK275106-OK275134.

## 3. Results

### 3.1. Isolation and Phylogenetic Diversity of Intestinal Fungi from the Three Coral Reef Fish

A total of 387 fungal isolates were recovered from the three fish intestines (240 isolates) and the seawater (147 isolates) in which the fish lived, based on the size, color, and other morphological observations of the fungus. After being sequenced (ITS sequences) and BLAST searched in GenBank, only one isolate shared 90.40% (low) similarity with its closest NCBI relative (*Absidi psychrophilia* strain SYM0202, its accession number is JN942684), which was excluded from our further analysis. The remaining 386 isolates shared 98–100% similarity to their closest NCBI relatives (Table 1).

Based on the fungal ITS-rDNA sequences, a phylogenetic tree of 29 different fungal species in 15 known genera is shown in (Figure 2), which was constructed using the neighbor-joining method. The 15 known genera recovered in this study mainly included *Aspergillus*, *Penicillium*, *Talaromyces*, *Aureobasidium*, *Cladosporium*, *Fusarium*, *Clonostachys, Myrothecium*, *Parengyodontium*, *Trichoderma, Hypocrea*, *Microsphaeropsi*, *Schizophyllum*, *Rigidoporus,* and *Cutaneotrichosporon*.

### 3.2. Dissimilarity of Fungal Communities in the Intestines of Three Coral Reef Fish and Seawater

The Bray-Curtis analysis showed distinct dissimilarity of the intestinal fungal communities between the three fish, which was from 41.2% to 71.4% (Table 2), indicating that the difference in the intestinal fungal communities between the three fish was obvious. After further analysis of intestinal fungi from the three fish species, it was found that many fungal species, such as *Aspergillus niger* and *Aspergillus pseudoglaucus*, were commonly recovered in all three fish species (Table 1). However, there were some distinctive fungal species distributed in different fish. For example, *Aspergillus medius*, *Aspergillus sydowii*, *Aureobasidium melanogenum*, *Aureobasidium pullulans,* and *Cladosporium halotolerans* were only isolated from *L. calcarifer*, while *Aspergillus ochraceopetaliformis*, *Microsphaeropsis arundinis*, *Myrothecium inundatum,* and *Rigidoporus vinctus* were unique to *L. argentimaculatus. Aspergillus restrictus*, *Cutaneotrichosporon jirovecii,* and *Penicillium sclerotiorum* were recovered only from *T. blochii* (Table 1).

The Bray–Curtis analysis showed a relatively high dissimilarity of fungal communities between fish intestinal fungi and surrounding seawater fungi, ranging from 66.7% to 81.8% (Table 2). This result indicated that the fungal community in three fish intestines differed from that in the seawater in which fish live.

### 3.3. Comparison of the Fungal Community in Fore-, Mid- and Hind-Intestinal Segments

Bray–Curtis analysis showed that the dissimilarities of the fungal communities in different segments of the same fish were from 27.3% to 100.0% (Table 3). The fungal isolates from different intestinal segments in the same fish showed distinct dissimilarity in different fish. The dissimilarities between the fore- and mid-intestinal segments of *L. calcarifer*, *L. argentimaculatus*, and *T. blochii* were 27.3%, 60.0%, and 100.0%, respectively. The dissimilarity between the mid- and hind-intestinal segments in *L. argentimaculatus* was 55.6%, yet the dissimilarity was 100.0% in *L. calcarifer* and *T. blochii*. The dissimilarity between fore- and hind-intestinal segments in *L. calcarifer*, *L. argentimaculatus,* and *T. blochii* was 71.4%, 33.3%, and 75.0%, respectively.

### 3.4. Analysis of the Antimicrobial Activity of the Tested Fungi

Thirty-seven fungal representative isolates (29 species) were tested against the three marine pathogenic fungi and three marine pathogenic bacteria. Nineteen isolates (51.4%) exhibited distinct antimicrobial activity against at least one pathogenic bacterium or fungus (Table 4). It is worth mentioning that isolate *A. pullulans* SCAU243 exhibited very strong antifungal activity against *A. versicolor*, while isolate *Schizophyllum commune* SCAU255 displayed moderate or strong antibacterial activity against four species of marine pathogenic microbial species (Table 4). In addition, *A. pseudoglaucus* SCAU239-1, *F. oxysporum* SCAU247-3, and *Hypocrea lixii* SCAU249 displayed moderate antimicrobial activity against three pathogenic microorganisms; and eight fungi showed antimicrobial activity against two pathogenic microorganisms.

## 4. Discussion

In this research, a total of 387 fungal isolates were recovered, and only 386 isolates were identified as 29 known fungal species, indicating the presence of relatively diverse and abundant fungal floras in the three coral reef fish intestines and their surrounding environments (seawater). One isolate showed 90.40% similarity with its closest relative, indicating that it may be a new species in the genus *Absidi.* The identification of the novel candidate still needs further investigation through morphological observation and multigene analysis. The 386 identified isolates belonged to Ascomycota and Basidiomycota (phylum level), while Ascomycota fungi were common both in the three coral reef fish intestines and in seawater (Table 1). These results are consistent with the observations in Tilapia and Bighead Carp intestines, deep oceans, deep-sea sediments, and seawater [19,41,42,43,44].

### 4.1. Comparison of Fungal Community in Environments and Intestines from the Three Fish

In general, Bray–Curtis analysis showed a distinct dissimilarity in the fungal species in the fish intestines of the three reef fish species (Table 2). Previous studies generally believe that the similarity of intestinal microbial colonization in fish is partly due to the fluctuation in the environmental habitat [18]. Among the environmental habitats, the microbial compositions of the water and diet are the decisive factors [45,46], suggesting that environmental factors were able to influence the composition of the intestinal microbiota of diseased fish, and their experimental results showed that the two most important environmental factors affecting the intestinal microbiota of “red operculum” diseased fish were ammonia concentration and water temperature. Coincidentally, all three fish species are carnivorous and live in the same environment in this study. Therefore, it was hypothesized that the same living environment might cause the three fish to have similar intestinal fungal communities. In addition, the reasons for the similarity of the intestinal fungi of the three fish should include the fact that only 1% of the total microbial community was culturable [47]. On the other hand, although they all live in the same environment in this experiment, and the effects of salinity, temperature, and microbial composition of seawater can be excluded, there is still a certain degree of dissimilarity of environmental conditions between them. Ref. [18] reported that the host could select different fungal species from the environment to colonize in the intestines. Besides surrounding environments, the intestinal fungal communities of the three coral reef fish species in the South China Sea may also be affected by potential factors such as the sex, size, life stages, feeding strategies, and potential prey of these fish species. For example, Li et al. revealed that the difference had occurred in the intestinal microbial communities between male and female wild largemouth bronze gudgeon [48]. Furthermore, [49] found that the intestinal microflora of Atlantic cod *Gadus morhua* differed depending on whether the fish were fed fish meal, fermented soy protein, or standard soy protein.

### 4.2. Comparison of Fungal Community in Fore-, Mid- and Hind-Intestinal Segments 

The dissimilarity of the fungal isolates in different segments of the same fish was from 27.3% to 100.0% (Table 3), suggesting that the diversity and distribution of the fungal community varied with different intestinal segments in the fish. Different intestinal segments of vertebrates, including fish, show various features, such as digestion and absorption, which lead to different microbial compositions in different intestinal segments [50,51]. For example, microbial communities in the intestines of adult honeybees and wild-caught adult *Penaeus monodon* differed significantly from the crop to the rectum, and the similarity between communities was significantly reduced, demonstrating a niche partitioning and compartment specificity [52,53]. However, in the present study, significant differences were observed only between the fore- and mid-intestines of *L. calcarifer* and between the three different intestinal segments of *T. blochii*. While no remarkable differences were observed for fungal communities in other segments of the intestine, which was comparable to the findings for bighead carp, where there were no remarkable differences in fungi between intestinal segments [19]. Among the intestinal fungi, we found a dramatic decrease in the number of yeasts in the hind-intestine in comparison to the fore- and mid-intestines, in addition to differences in the distribution of *Trichoderma* spp. and *Talaromyces* spp. In a survey of tilapia and bighead carp, it was also found that the hind-intestines showed a significant reduction in pathogenicity and saprotrophicity compared with the fore-intestine [19]. These results indicated that the distribution of the fungi in the fish intestine was associated with the physiological functions of various intestinal segments except for the specificity of the host species.

### 4.3. Potential Antimicrobial Properties of Fungi from Fish Intestine and Seawater

Thirty-seven fungal representatives from the intestine of three coral reef fish and seawater were tested for antimicrobial activity against six pathogenic bacteria and fungi. After initial screening, more than half of the fungal isolates (51.4%) showed significant antimicrobial activities in the experiment, indicating that fungi from the fish intestines or seawater were an excellent source for the extraction of natural products with great bioactivity [54,55]. *A. versicolor* is a pathogen [38] that can be inhibited by a variety of fungi from fish intestines and seawater in this investigation (Table 4). It is widely distributed in humid indoor environments and causes not only remarkable lethal activity to brine shrimp but also respiratory and pulmonary diseases in humans; moreover, it accounts for 5.8% of the fungi that cause gray nails in recent statistics [56,57]. In this assay, five isolates of experimental fungi demonstrated high inhibitory activity against *A. versicolor* (Table 4), and the inhibitory activity of *A. pullulans* SCAU243 far exceeded that of other fungi.

Among the fungal isolates with broad antimicrobial activity, *S. commune* SCAU255 exhibited extensive and high-intensity antibacterial activity [58]. As a widespread edible fungus with distribution on almost every continent except Antarctica, *S. commune* produces the neutral extracellular polysaccharide lytic polysaccharide, consisting of a 1,6-β-d-glucosyl side group and a 1,3-β-d-linked glucose residue backbone, with remarkable anticancer and antitumor activity and good immunomodulatory activity, yet it is less frequently found in seawater [59,60]. In previous studies, isolates of *S. commune* found in marine sediments showed biological activity inconsistent with continental isolates, and it was hypothesized that isolates from marine sources might have unique biological characteristics that assist them in adapting to extreme seafloor environments [61].

For the remaining isolates with antimicrobial activities, except for some species of *Talaromyces* and *Aspergillus* that exhibited relatively high antibacterial and antifungal activities, *Microsphaeropsis arundinis* SCAU250 and *Penicillium sclerotiorum* SCAU253 only had high antibacterial activities. Interestingly, three fungal isolates of *Fusarium oxysporum* and two isolates of *Aspergillus pseudoglaucus* demonstrated different antibacterial and antifungal activities. These findings illustrate that different isolates of *F. oxysporum* and *A. pseudoglaucus* may have relatively high variability and differ in morphology, biological activity, and pathogenicity. The reason for this may be attributed to diverse karyotypes among isolates in the same environment [62]. As a fish pathogenic fungus, *F. oxysporum* has been found in many fish in recent years, and it poses a great danger to fish health [63,64,65]. Different countermeasures, such as different biological activity screening modes, and different application directions, etc., must be taken regarding different isolates. 

## 5. Conclusions

In conclusion, fungal diversity in the intestine of three species of coral reef fish and seawater was investigated using culturable methods and ITS sequences. Both fish intestine and seawater fungi showed high diversity. There were similarities in the intestinal fungal communities between the three fish and seawater, which verified the influence of seawater on the intestinal microbial colonization of fish. In the comparison of the variability of different segments of fish intestinal fungi, significant differences were found only between the fore- and mid-intestines of *Lates calcarifer* and between the three different intestinal segments of *Trachinotus blochii*. However, the number of culturable microbes is quite different from the actual number of microbes in the environment and fish intestine. Relying on culture technology alone cannot fully reflect the true situation of marine microbial diversity and may lead to the omission of microbial resources with potential application value, which is a shortcoming of our study. The combination of culturable and high-throughput sequencing will be the direction of future research on fungal diversity.

## Figures and Tables

**Figure 1 jof-09-00613-f001:**
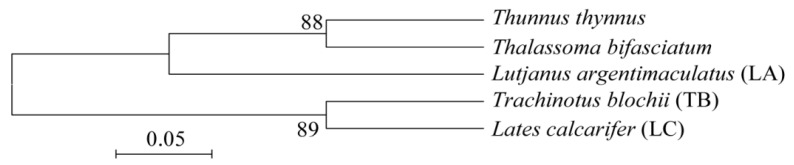
Neighbor-Joining phylogenetic tree of the cytochrome c oxidase subunit I (COI) gene of the three coral reef fish (*Lates calcarifer, Trachinotus blochii, and Lutjanus argentimaculatus)* with the maximum composite likelihood and bootstrap method (Q = 1000) was rooted by *Thunnus thynnus, Thalassoma bifasciatum,* and constructed with Mega 6.0. The clone sequences of the fish were downloaded from the National Center for Biotechnology Information (NCBI, Bethesda, MD, USA)*,* and the corresponding locus tags were KT352986.1, HM379862.1, GU673901.1, GU673843.1, and JX983354.1.

**Figure 2 jof-09-00613-f002:**
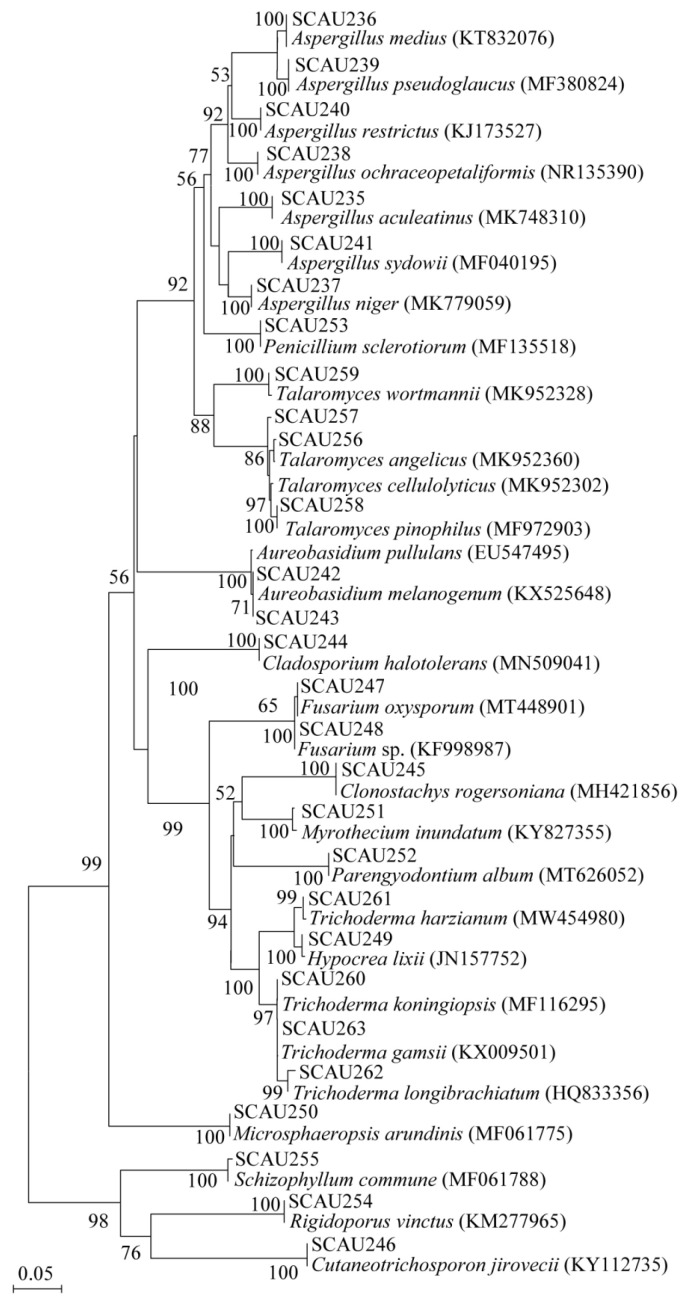
Neighbor-joining phylogenetic tree from analysis of the internal transcribed spacer (ITS) sequences of culturable fungi isolated from three coral reef fish intestines and seawater. The number on the node represents the percentage of bootstrapping support based on a neighbor-joining analysis of 1000 resampled datasets. Only values of >50% are shown. Scale bar: 0.05 substitutions per nucleotide position.

**Table 1 jof-09-00613-t001:** Fungi isolated from three fish intestines and environmental seawater as identified by internal transcribed spacer (ITS) sequences.

Fungal Species	Representative Isolates (Accession Number in GenBank)	Similarity of ITS Sequence	Number of Fungal Isolates			
			LC	LA	TB	SW
*Aspergillus aculeatinus*	SCAU235 (OK275106)	99.6%				
*A. medius*	SCAU236 (OK275107)	99.4%	4 ± 1			
*A. niger*	SCAU237 (OK275108)	100.0%	1 ± 1	9 ± 2	5 ± 2	7 ± 1
*A. ochraceopetaliformis*	SCAU238 (OK275109)	99.8%		3 ± 2		
*A. pseudoglaucus*	SCAU239 (OK275110)	99.0%	2 ± 1	2 ± 0	2 ± 1	
*A. restrictus*	SCAU240 (OK275111)	99.1%			1 ± 1	
*A. sydowii*	SCAU241 (OK275112)	99.3%	10 ± 3		4 ± 1	
*Aureobasidium melanogenum*	SCAU242 (OK275113)	99.6%	1 ± 0			3 ± 1
*A. pullulans*	SCAU243 (OK275114)	99.8%	2 ± 1		4 ± 1	5 ± 2
*Cladosporium halotolerans*	SCAU244 (OK275115)	100.0%	3 ± 1		4 ± 3	2 ± 1
*Clonostachys rogersoniana*	SCAU245 (OK275116)	99.6%				3 ± 1
*Cutaneotrichosporon jirovecii*	SCAU246 (OK275117)	99.4%			1 ± 0	
*Fusarium oxysporum*	SCAU247 (OK275118)	99.8%				2 ± 0
*Fusarium* sp.	SCAU248 (OK275119)	99.8%				2 ± 1
*Hypocrea lixii*	SCAU249 (OK275120)	99.3%				3 ± 2
*Microsphaeropsis arundinis*	SCAU250 (OK275121)	99.0%				2 ± 1
*Myrothecium inundatum*	SCAU251 (OK275122)	99.67%		2 ± 1		
*Parengyodontium album*	SCAU252 (OK275123)	99.6%		2 ± 1	2 ± 1	
*Penicillium sclerotiorum*	SCAU253 (OK275124)	99.5%			2 ± 0	
*Rigidoporus vinctus*	SCAU254 (OK275125)	99.5%		1 ± 0		2 ± 1
*Schizophyllum commune*	SCAU255 (OK275126)	99.7%				2 ± 1
*Talaromyces angelicus*	SCAU256 (OK275127)	98.4%				1 ± 0
*T. cellulolyticus*	SCAU257 (OK275128)	98.9%	2 ± 1			
*T. pinophilus*	SCAU258 (OK275129)	99. 5%				3 ± 1
*T. wortmannii*	SCAU259 (OK275130)	99.5%				3 ± 1
*Trichoderma gamsii*	SCAU260 (OK275131)	99.7%				3 ± 1
*T. harzianum*	SCAU261 (OK275132)	100.0%				1 ± 0
*T. koningiopsis*	SCAU262 (OK275133)	99.1%				2 ± 1
*T. longibrachiatum*	SCAU263 (OK275134)	99.7%		6 ± 2	2 ± 1	3 ± 1

LC *Lates calcarifer*, LA *Lutjanus argentimaculatus*, and TB *Trachinotus blochii*. “SW” refers to the surrounding seawater.

**Table 2 jof-09-00613-t002:** Dissimilarity of fungal communities in guts of three species of coral reef fish and environmental seawater.

	LC	LA	TB	SW
LC	/	71.4%	41.2%	66.7%
LA	71.4%	/	60.0%	81.8%
TB	41.2%	60.0%	/	76.0%
SW	66.7%	81.8%	76.0%	/

LC *Lates calcarifer*, LA *Lutjanus argentimaculatus*, and TB *Trachinotus blochii*. “SW” refers to the surrounding seawater.

**Table 3 jof-09-00613-t003:** The dissimilarity of fungal communities in fore-, mid-, and hind-intestinal segments of the same fish.

	LC-F	LC-M	LC-H	LA-F	LA-M	LA-H	TB-F	TB-M	TB-H
LC-F	/	27.3%	71.4%	100.0%	60.0%	100.0%	75.0%	66.7%	100.0%
LC-M	27.3%	/	100.0%	81.8%	45.5%	80.0%	77.8%	71.4%	81.8%
LC-H	71.4%	100.0%	/	100.0%	100.0%	100.0%	60.0%	100.0%	100.0%
LA-F	100.0%	81.8%	100.0%	/	60.0%	33.3%	75.0%	100.0%	20.0%
LA-M	60.0%	45.5%	100.0%	60.0%	/	55.6%	75.0%	66.7%	60.0%
LA-H	100.0%	80.0%	100.0%	33.3%	55.6%	/	71.4%	100.0%	33.3%
TB-F	75.0%	77.8%	60.0%	75.0%	75.0%	71.4%	/	100.0%	75.0%
TB-M	66.7%	71.4%	100.0%	100.0%	66.7%	100.0%	100.0%	/	100.0%
TB-H	100.0%	81.8%	100.0%	20.0%	60.0%	33.3%	75.0%	100.0%	/

LC *Lates calcarifer*, LA *Lutjanus argentimaculatus*, and TB *Trachinotus blochii*. F fore-intestinal segments, M mid-intestinal segments, and H hind-intestinal segments.

**Table 4 jof-09-00613-t004:** Antimicrobial activities of fungi in three fish intestines and environmental seawater.

Fungal Species	Representative	Diameter of the Growth Inhibition Zone (mm)					
	Isolates	PP	VA	ML	AV	AS	PC
*Aspergillus* *ochraceopetaliformis*	SCAU238	/	11.3 ± 5.3	/	/	/	/
*A*. *pseudoglaucus*	SCAU239-1	/	10.8 ± 0.3	/	/	19.0 ± 2.0	17.0 ± 1.0
*A*. *pseudoglaucus*	SCAU239-2	12 ± 1.0	/	/	20.0 ± 2.3	/	/
*A*. *sydowii*	SCAU241	/	/	/	20.5 ± 0.5	/	/
*Aureobasidium melanogenum*	SCAU242	13.5 ± 0.5	13.5 ± 1.5	/	/	/	/
*A. pullulans*	SCAU243	/	/	/	36.0 ± 1.0	/	/
*Cladosporium h* *alotolerans halo tolerans*	SCAU244	11.5 ± 0.5	/	/	/	17.5 ± 0.5	/
*Fusarium oxysporum*	SCAU247-1	/	/	/	/	/	17.0 ± 1.0
*F*. *oxysporum*	SCAU247-2	14.5 ± 2.5	/	/	/	/	/
*F*. *oxysporum*	SCAU247-3	14.5 ± 0.5	15.5 ± 3.5	/	/	/	20.5 ± 0.5
*Hypocrea lixii*	SCAU249	12.0 ± 0.3	13.0 ± 1.0	18.0 ± 4.0	/	/	/
*Microsphaeropsis arundinis*	SCAU250	/	/	/	22.5 ± 0.5	/	17.5 ± 1.5
*Parengyodontium album*	SCAU252	13.5 ± 1.5	13.5 ± 2.5	/	/	/	/
*Penicillium sclerotiorum*	SCAU253	/	/	/	/	15.5 ± 0.5	19.5 ± 2.5
*Schizophyllum commune*	SCAU255	12.5 ± 0.5	13.3 ± 2.7	13.0 ± 1.0	/	/	26.5 ± 3.5
*Talaromyces cellulolyticus*	SCAU257	/	/	/	17.8 ± 2.8	/	12.5 ± 0.5
*T*. *wortmannii*	SCAU259	/	/	/	26.0 ± 4.0	/	18.5 ± 1.5
*T. gamsii*	SCAU260	/	8.3 ± 3.9	/	/	/	/
*T*. *harzianum*	SCAU261	/	/	20.5 ± 5.5	/	/	/

PP *Pseudoaltermonas piscida*, VA *Vibrio alginolyticus*, ML *Micrococcus luteus*, AV *Aspergillus versicolor*, AS *A. sydowii,* and PC *Penicillium citrinum*. The experiment in the inhibition circle size differences manifest the degree of inhibition activity (Each trial was conducted three times): Zone of inhibition circle less than or equal to 10 mm for weak activity; ranged between 10 and 20 mm for moderate activity; greater than 20 mm for strong activity; “/”: no traces or no antagonistic effects were detected.

## Data Availability

The ITS sequences of the 29 representative fungal isolates obtained in this experiment were registered in GenBank. Accession numbers are OK275106-OK275134.

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
