# Peer review of "Diversity and Antimicrobial Activity of Intestinal Fungi from Three Species of Coral Reef Fish"

_jof, 2023, doi:10.3390/jof9060613_

Round 1

Reviewer 1 Report

The manuscript submitted by Liao et al entitled “Diversity and antimicrobial activity of intestinal fungi from three species of coral reef fishes” mainly discuss the fungal isolates from fish and its intestinal microbiota and its related. The manuscript is well written and covered the related topics with respect to the scope of this article in an understandable way. However, I suggest the author to address the following comments.

Line 139: Why sand was added while grinding? Just add the reason

Line 149-150: Can the authors mention the approximate days took to see the morphology of isolates?

Lines 160, 163, 165: Is the mentioned r/min , rpm or rcf? Mention clearly

Lines 235, 269: The fish names should be written in italics and space between Trachinotus and blochii

Line 416: Mention atleast 2-3 countermeasures

Good luck with submission.

Author Response

Dear reviewer,

Journal of Fungi,

On behalf of my co-authors, I would like to submit the revised manuscript entitled “Diversity and antimicrobial activity of intestinal fungi from three species of coral reef fish” (jof-2345361) to Journal of Fungi.

The authors are grateful to the reviewers for their valuable comments and time, and all comments are responded carefully point by point.

**********************************

To Reviewer :

Details of the comments are as follows:

Abstract

  1. Line 139: Why sand was added while grinding? Just add the reason

Response: Thank you for your suggestion, which we had explained in the revised manuscript: the sand serves to make the sample fully ground.

  1. Line 149-150: Can the authors mention the approximate days took to see the morphology of isolates?

Response: Thank you for your good advice, which we had detailed in the revised manuscript: The approximate days took to see the morphology of isolates was 5-7 days.

  1. Lines 160, 163, 165: Is the mentioned r/min, rpm or rcf? Mention clearly

Response: Thank you for your suggestion, I have added clarification to the question and the r/min mentioned in the article refers to revolutions per minute.

  1. Lines 235, 269: The fish names should be written in italics and space between Trachinotus and blochii

Response: Thanks to your correction, the fish names had been written in italics and space between Trachinotus and blochii.

  1. Line 416: Mention at least 2-3 countermeasures

Response: Thank you for your good advice. The following sentence had been added into the revised manuscript.

Different countermeasures, such as different biological activity screening modes, and different application directions, etc., must be taken in regarding different isolates.

**********************************

All revised contents in text were highlighted in red in the revised manuscript.

Thank you very much for your consideration of our manuscript for potential publication. We look forward to hearing from you soon.

Best Regards.

Sincerely yours,

Dr. Xiaoyong Zhang

University Joint Laboratory of Guangdong Province, Hong Kong and Macao Region on Marine Bioresource Conservation and Exploitation, College of Marine Sciences, South China Agricultural University, Guangzhou 510642, China

E-mail: zhangxiaoyong@scau.edu.cn

Reviewer 2 Report

In this manuscript, the authors describe the diversity and antimicrobial activity of intestinal fungi from three species of coral reef fish. The manuscript is interesting and could be accepted after the modification of text in the introduction and description sections.

The introduction is necessary to provide more specific and updated relevant reports from public literature.

The discussion section is too long but without relevant input. A cutting-edge discussion is required to establish current findings. 

Also, the references need to be updated with more relevant reports.

Author Response

Dear reviewer,

Journal of Fungi,

On behalf of my co-authors, I would like to submit the revised manuscript entitled “Diversity and antimicrobial activity of intestinal fungi from three species of coral reef fish” (jof-2345361) to Journal of Fungi.

The authors are grateful to the reviewers for their valuable comments and time, and all comments are responded carefully point by point.

**********************************

To Reviewer :

The text of the introduction and description sections of the manuscript have been revised in line with the reviewers' suggestions. A more specific and updated version of the relevant report has been consulted and made available. And some elements of the discussion section have been removed and relevant input added. Details of the comments are as follows:

Introduction

  1. The introduction is necessary to provide more specific and updated relevant reports from public literature.

Response: Thank you for your good advice, we have reviewed the relevant literature and have now included the relevant content in the Part of Introduction. Which were as follows.

(1)A recent study revealed that the fungal communities were not only related to fish species specificity but also to the respective physiological functions of different intestinal segments [19]. Among fungi, yeasts have been recognized as part of the normal microbiota of fish. Genus Rhodotorula was relatively frequent in both freshwater and marine fish, and genus Debaryomyces were dominant in rainbow trout [22]. Caruffo et al. reported that many yeasts isolated from the gut of fish could be potential probiotics, reducing the mortality associated with Vibrio anguillarum challenge [23].

(2)These coral reef fish generally have a complex life history. Spawning and larval settlement occurred in the marine areas of river mouths, and then juvenile fish migrate upstream to fresh water where they grew and became mature as males [28]. They exhibited a relatively high trophic status in the food chain and played a crucial role in maintaining the ecological balance of coral reef systems. They fed on fish and crustaceans, but consumed a variety of algae as well [2].

(3)The information on fungal diversity and distribution in different intestinal segments of coral reef fish is very scarce and important, which provides the baseline data for gut microbiota in coral reef fish from the South China Sea, and a reference for the study on gut microbiota in other marine organisms.

(4)In addition, 37 representative fungal isolates were selected for screening their antimicrobial activity against six marine pathogenic microorganisms, which can check the potential of these fungi from different intestinal segments in the production of bioactive molecules against pathogenic microorganisms and provide a good resource for our subsequent screening of marine microbial active substances.

The following references were added into the revised manuscript.

[2] Gao, Y.M.; Zou, K.S.; Zhou, L.; Huang, X.D.; Li, Y.Y.; Gao, X.Y.; Chen, X.; Zhang, X.Y. Deep insights into gut microbiota in four carnivorous coral reef fish from the south china sea. Microorganisms 2020, 8, doi: 10.3390/microorganisms8030426.

[19] Zhou, L.; Han, Y.; Wang, D.; Li, Y.; Huang, X.; He, A. Comparison of fungal community composition within different intestinal segments of tilapia and bighead carp. Journal of Oceanology and Limnology 2021, 39, 1961-1971, doi: 10.1007/s00343-020-0214-3.

[22] Gatesoupe, F.J. Live yeasts in the gut: natural occurrence, dietary introduction, and their effects on fish health and development. Aquaculture 2007, 267, 20-30, doi: 10.1016/j.aquaculture.2007.01.005.

[23] Caruffo, M.; Navarrete, N.; Salgado, O.; Díaz, A.; López, P.; García, K.; Feijóo, C.G.; Navarrete, P. Potential probiotic yeasts isolated from the fish gut protect zebrafish (Danio rerio) from a vibrio anguillarum challenge. Frontiers in Microbiology 2015, 6, doi: 10.3389/fmicb.2015.01093.

[28] McCulloch, M.; Cappo, M.; Aumend, J.; Müller, W. Tracing the life history of individual barramundi using laser ablation mc-icp-ms sr-isotopic and sr/ba ratios in otoliths. Marine and Freshwater Research 2005, 56, 637, doi: 10.1071/MF04184.

Discussion

  1. The discussion section is too long but without relevant input. A cutting-edge discussion is required to establish current findings.

Response: Thank you for your good advice, we have removed some of the content from the discussion section and we have reviewed the relevant literature and added newer and more cutting edge relevant content. The contents of which are as follows.

The following content and the corresponding references have been removed:

Some of the fungi in the seawater were also present in the fish intestine, including Aspergillus spp., Aureobasidium spp., Cladosporium spp., and Trichoderma spp.. Aspergillus species were the most common and diverse fungi [44]. As one of the most plentiful fungi in nature, most Aspergillus species can be widely distributed on grains, in air, soil, and various organic materials, and in marine environments. They have been found in the intestines of many mammals, even including humans. As fungi derived from marine origins, Cladosporium and Trichoderma species are widely found in sediments, seagrass, sponges, and different fish species.

Yet, Aureobasidium is a black yeast-like fungus, and some strains produce various polysaccharides, including β-glucan, which are known to exert immunostimulatory effects. It is present in the intestinal tract of many freshwater fish as symbiotrophic fungi, such tilapia (Oreochromis mossambicus) and bighead carp (Aristichthys nobilis) . And Aureobasidium had also been found in the intestine of marine fish. For example, found the presence of Aureobasidium in carnivorous salmonids. Therefore Aureobasidium is presumed to be autochthonous (able to colonize in the gut) . discovered that A. pullulans strains from different marine environments had different physiological characteristics including secreting cellulose extracellular amylase, and killer toxin, yielding lipase and cellulase. These functions may accelerate the decomposition of organic matter in fish. In addition, in the present experiment, A. pullulans was found to have an extremely high-level antipathogenic fungal activity, and A. pullulans could produce siderophores to bind iron and take up the siderophore-iron complex for their growth. We hypothesized that Aureobasidium spp. entered fish intestinal tract because of the above functions.

The following is newly consulted and relevant content has been added to the article:

(1)Besides surrounding environments, the intestinal fungal communities of the three coral reef fish species in the South China Sea may also be affected by potential factors such as the sex, size, life stages, feeding strategies, and potential prey of these fish species. For example, Li et al. revealed that the difference had occurred in the intestinal microbial communities between male and female wild largemouth broze gudgeon [48] . Furthermore, [49] found that the intestinal microflora of Atlantic cod Gadus morhua differed depending on whether the fish were fed fish meal, fermented soy protein or standard soy protein.

The following references were added into the revised manuscript.

[48] Li, X.; Yan, Q.; Ringø, E.; Wu, X.; He, Y.; Yang, D. The influence of weight and gender on intestinal bacterial community of wild largemouth bronze gudgeon (Coreius guichenoti, 1874). Bmc Microbiology 2016, 16, doi: 10.1186/s12866-016-0809-1.

[49] Ringø, E.; Sperstad, S.; Myklebust, R.; Refstie, S.; Krogdahl, Å. Characterisation of the microbiota associated with intestine of atlantic cod (Gadus morhua l.): The effect of fish meal, standard soybean meal and a bioprocessed soybean meal. Aquaculture 2006, 261, 829-841, doi: 10.1016/j.aquaculture.2006.06.030.

**********************************

All revised contents in text were highlighted in red in the revised manuscript.

Thank you very much for your consideration of our manuscript for potential publication. We look forward to hearing from you soon.

Best Regards.

Sincerely yours,

Dr. Xiaoyong Zhang

University Joint Laboratory of Guangdong Province, Hong Kong and Macao Region on Marine Bioresource Conservation and Exploitation, College of Marine Sciences, South China Agricultural University, Guangzhou 510642, China

E-mail: zhangxiaoyong@scau.edu.cn

Reviewer 3 Report

This manuscript used a culturable method and the phylogenetic analysis of internal transcribed spacer (ITS) sequences to examine the diversity and antimicrobial activity of intestinal fungi from three coral reef fish species in the South China Sea.  The paper is interesting and potentially useful, as it demonstrated that more than half of the fungal isolates (51.4%) showed significant antimicrobial activities against marine pathogenic microorganisms, and made an attempt to suggest that the distribution of fungi in fish intestines may be related to the physiological functions of different intestinal segments and the fungal colonization may be influenced by fish surrounding environments.  I think that the comparison of the diversity and antimicrobial activity of culturable fungi from different intestinal segments of multiple species and their living environments, in a phylogenetic analysis of ITS sequences framework, is an interesting advantage of this paper in relation to others that deal with similar topics.  There are some justifications the authors need to include, which will enrich the content of the research while clarifying the selection and implementation of the approaches used.  The specific comments are:

Abstract

-        (1)  Page 1 Line 16.  blochii and” to “blochii, and”.

-        (2)  Page 1 Line 19.  “fungal community” to “fungal communities”.

-        (3)  Page 1 Line 19.  “verified the” to “verified that the”.

-        (4)  Page 1 Line 22.  “fore- and mid-intestine” to “fore- and mid-intestines”.

-        (5)  Page 1 Line 22.  “fishes intestines” to “fishes’ intestines”.

-        (6)  Page 1 Line 23.  “various intestinal segment” to “various intestinal segments”.

-        (7)  Page 1 Line 25.  “microorganisms” to “microorganism”.

-        (8)  Page 1 Line 28.  “further increasing” to “further increased”.

Introduction

-        (9)  Page 1 Line 33.  “for great” to “for their great”.

-        (10)  Page 1 Line 33.  “ecological and” to “ecological, and”.

-        (11)  Page 1 Line 35.  “habitat” to “habitats”.

-        (12)  Page 1 Line 43.  “the complicated interactions” to “the complex interactions”.

-        (13)  Page 2 Line 46.  For “among the different species”, do you mean “among the different fish species”?

-        (14)  Page 2 Lines 46-47 (and others throughout the manuscript).  “golden pompano” to “Golden Pompano”.  I would suggest that the common names of fish species be capitalized, following the usage in the 7th edition of Common and Scientific Names of Fishes from the United States, Canada, and Mexico (AFS Special Publication 34; 2013).

-        (15)  Page 2 Line 47.  “Spirochaetes and” to “Spirochaetes, and”.

-        (16)  Page 2 Line 48.  “barramundi” to “Barramundi”.

-        (17)  Page 2 Line 50.  Pseudomonas” to “and Pseudomonas”.

-        (18)  Page 2 Lines 50-51.  “mangrove red snapper” to “Mangrove Red Snapper”.

-        (19)  Page 2 Line 52.  Pantoea and” to “Pantoea, and”.

-        (20)  Page 2 Line 54.  Firmicutes and” to “Firmicutes, and”.

-        (21)  Page 2 Line 58.  “able to colonize” to “being able to colonize”.

-        (22)  Page 2 Line 60.  “they promote” to “as they promote”.

-        (23)  Page 2 Line 75.  “fish, however” to “fish. However”.

-        (24)  Page 2 Line 77.  “in situ” should be italicized.

-        (25)  Page 2 Line 77.  “[24] and” to “[24], and”.

-        (26)  Page 2 Line 80.  “golden permits (T. blochii) and” to “Golden Permit (T. blochii), and”.

-        (27)  Page 2 Line 81.  “from the South China Sea” to “in the South China Sea”.

-        (28)  Page 2 Line 83.  “of three coral reef fishes” to “of the three coral reef fishes”.

-        (29)  Page 2 Line 88.  “dissimilarity of the fungal community” to “dissimilarity of the fungal communities”.

-        (30)  Some additional information such as brief background on the life history, feeding strategies, and potential prey of the three coral reef fish species in the South China Sea, which may be related to their intestinal fungal communities, would be helpful in the Introduction section to set the stage.  I would also suggest emphasizing somewhere in the Introduction section the significance of the comparison of the diversity and antimicrobial activity of culturable fungi from different intestinal segments of multiple species and their surrounding environments in a phylogenetic analysis of ITS sequences framework, because these approaches might help people from broader field find your work useful rather than people just working on the particular species or area being interested in it.

Materials and Methods

-        (31)  Page 3 Line 98.  “an average annual temperature” to “an annual average temperature”.

-        (32)  Page 3 Line 99.  “The maximum temperature of the sea surface is” to “The maximum sea surface temperature (SST) is”.  I would suggest using “sea surface temperature”, as SST is a commonly used variable in marine science.

-        (33)  Page 3 Line 102.  “TB3) and” to “TB3), and”.

-        (34)  Page 3 Lines 104-106.  The intestinal fungal communities of the three coral reef fish species may be affected by the sex, size, and life stages of fish samples collected.  Have you considered that?  I would suggest adding some information on the sex, size, and life stages of fish samples collected in this study.

-        (35)  Page 3 Line 105.  “each species fish” to “each fish species”.

-        (36)  Page 3 Line 105.  I would suggest using “nine” rather than “9” when the number is less than ten.

-        (37)  Page 3 Line 106.  “TB3 and” to “TB3, and”.

-        (38)  Page 4 Line 147.  “agar) and SYA” to “agar), and SYA”.

-        (39)  Page 4 Line 151.  “agar and” to “agar, and”.

-        (40)  Page 4 Line 157.  “taken into in a” to “taken into a”.

-        (41)  Page 4 Line 159.  “watered bath at 65 °C” to “and the mixture was put in a water bath at 65 °C”.

-        (42)  Page 4 Line 159.  “min, centrifuged at 12000” to “min and then centrifuged at 12,000”.

-        (43)  Page 4 Lines 162-163.  “centrifuged it at 12000 r/min for 5min” to “and centrifuged at 12,000 r/min for 5 min”.

-        (44)  Page 4 Line 163.  “an equal volume” to “and an equal volume”.

-        (45)  Page 4 Line 164.  “tube, the centrifuge tube” to “tube. The centrifuge tube”.

-        (46)  Page 4 Line 165.  “12000” to “12,000”.

-        (47)  Page 4 Line 189.  “1000” to “1,000”.

-        (48)  Page 5 Line 195.  “PP) and” to “PP), and”.

-        (49)  Page 5 Line 195.  “[34], and” to “[34]; and”.

-        (50)  Page 5 Line 197.  “AS) and” to “AS), and”.

-        (51)  Page 5 Line 198.  “7 days” to “seven days”.

-        (52)  Page 5 Line 207.  “The abundance/species richness” to “The species abundance/richness”.

-        (53)  Page 5 Lines 207-209.  There is little explanation as to why the Bray-Curtis dissimilarity statistic was chosen in this study.  There are some other statistical approaches that have been used in ecology and environmental science when it comes to analyzing species’ relative abundance and making pairwise comparisons.  However, there is little justification as to why the authors only chose this method.  I would suggest adding a brief explanation to justify why this approach and not other also commonly used ones was chosen.  Given the potential of statistical analyses to influence the assessment of species’ relative abundance and compositional comparisons, it would be better to critically assess why some approaches are chosen and their potential and weaknesses for the available data and specific environment considered.

-        (54)  Page 5 Lines 207-209.  Did you use raw or transformed values of the species abundance and richness data when applying the statistical analyses?  It would be better to specify a little more even though data transformation may not be necessary for some situations, because it is an important step to make sure the variables meet the underlying assumptions of the algorithms before conducting any statistical analyses.  Statistical approaches with different mechanisms may have different underlying assumptions of normality, linearity or multicollinearity, and some variables may need to be transformed to meet specific assumptions.

Results

-        (55)  Page 7 Line 253.  “community between” to “communities between”.

-        (56)  Page 7 Line 254.  “the difference in…was not obvious” is not consistent with the “distinct dissimilarity” in Line 252.

-        (57)  Page 7 Line 255.  “from three fish species” to “from the three fish species”.

-        (58)  Page 7 Line 259.  pullulans and” to “pullulans, and”.

-        (59)  Page 7 Line 261.  inundatum and” to “inundatum, and”.

-        (60)  Page 7 Line 262.  jirovecii and” to “jirovecii, and”.

-        (61)  Page 8 Line 272.  “fungal community” to “fungal communities”.

-        (62)  Page 8 Line 277.  “fungal community” to “fungal communities”.

-        (63)  Page 8 Line 281.  “60.0% and” to “60.0%, and”.

-        (64)  Page 8 Line 284.  argentimaculatus and” to “argentimaculatus, and”.

-        (65)  Page 8 Line 285.  “33.3% and” to “33.3%, and”.

-        (66)  Page 9 Line 298.  “SCAU247-3 and” to “SCAU247-3, and”.

Discussion

-        (67)  Page 9 Line 310.  “closest relative indicating” to “closest relative, indicating”.

-        (68)  Page 9 Line 313.  “isolates were belonged” to “isolates belonged”.

-        (69)  Page 9 Line 315.  “and seawater” to “and in seawater”.

-        (70)  Page 9 Line 316.  “tilapia and bighead carp” to “Tilapia and Bighead Carp”.

-        (71)  Page 9 Line 319.  “a distinct similarity” is not consistent with the “distinct dissimilarity” in Line 252.

-        (72)  Page 10 Line 323.  “composition” to “compositions”.

-        (73)  Page 10 Line 323.  “factors [41-42]suggested” to “factors. [41-42] suggested”.

-        (74)  Page 10 Line 330.  “fungal community” to “fungal communities”.

-        (75)  Page 10 Line 335.  “dissimilarity between” to “dissimilarity of environmental conditions between”.

-        (76)  Page 10 Line 335.  “them [18]. reported” to “them. [18] reported”.

-        (77)  Page 10 Line 340.  “most species” to “most Aspergillus species”.

-        (78)  Page 10 Line 341.  “materials and” to “materials, and”.

-        (79)  Page 10 Line 343.  “sediment” to “sediments”.

-        (80)  Page 10 Line 365.  “leads” to “lead”.

-        (81)  Page 10 Line 366.  “microbial community in the intestine” to “microbial communities in the intestines”.

-        (82)  Page 10 Lines 366-367.  “honeybees, wild-caught” to “honeybees and wild-caught”.

-        (83)  Page 10 Lines 367-368.  “the similarity between community” to “and the similarity between communities”.

-        (84)  Page 10 Line 370.  “the fore- and mid-intestine” to “the fore- and mid-intestines”.

-        (85)  Page 11 Line 376.  “the fore- and mid-intestine” to “the fore- and mid-intestines”.

-        (86)  Page 11 Line 390.  “and not only causes remarkable” to “and causes not only remarkable”.

-        (87)  Page 11 Line 392.  “5.8% of the percentage of fungi that” to “5.8% of the fungi that”.

-        (88)  Besides surrounding environments, the intestinal fungal communities of the three coral reef fish species in the South China Sea may also be affected by potential factors such as the sex, size, life stages, feeding strategies, and potential prey of these fish species.  I would suggest acknowledging in the Discussion section these additional factors based on literature and their potential influence on the intestinal fungal communities of coral reef fish in the study area.

Conclusions

-        (89)  Page 11 Line 420.  “abundant diversity” to “high diversity”.

-        (90)  Page 11 Line 422.  “In a comparison of the” to “In the comparison of the”.

-        (91)  Page 11 Line 423.  “of intestinal fungi” to “of fish intestinal fungi”.

-        (92)  Page 11 Line 424.  “the fore and mid-intestine” to “the fore- and mid-intestines”.

Tables and Figures

-        (93)  Page 3 Figure 1 Line 119.  Lutjanus” to “and Lutjanus”.

-        (94)  Page 3 Figure 1 Lines 120-121.  Species name should be italicized.

-        (95)  Page 3 Figure 1 Line 122.  “from NCBI and” to “from the National Center for Biotechnology Information (NCBI, Bethesda, MD, USA) and”.

-        (96)  Page 3 Figure 1 Line 123.  “JX983354.1” to “and JX983354.1”.

-        (97)  Page 6 Table 1 Line 233.  “by ITS sequences” to “by internal transcribed spacer (ITS) sequences”.

-        (98)  Page 6 Table 1 Line 235.  Species name should be italicized.

-        (99)  Page 7 Figure 2 Line 246.  “the ITS sequences” to “the internal transcribed spacer (ITS) sequences”.

-        (100)  Page 7 Figure 2 Line 248.  “based on based on” is duplicated.

-        (101)  Page 8 Table 2 Line 269.  Species name should be italicized.

-        (102)  Page 8 Table 3 Line 288.  Species name should be italicized.

-        (103)  Page 9 Table 4.  I would suggest specifying in the table caption or as a footnote what “PP”, “VA”, “ML”, “AV”, “AS”, and “PC” represent.  A good table or figure caption should make the table or figure understandable without reference to the main text.

I would suggest minor editing of English language.

Author Response

Dear reviewer,

Journal of Fungi,

On behalf of my co-authors, I would like to submit the revised manuscript entitled “Diversity and antimicrobial activity of intestinal fungi from three species of coral reef fish” (jof-2345361) to Journal of Fungi.

The authors are grateful to the reviewers for their valuable comments and time, and all comments are responded carefully point by point.

**********************************

To Reviewer :

Details of the comments are as follows:

Abstract

  • Page 1 Line 16.  “blochii and” to “blochii, and”.

Response: Already modified “blochii and” to “blochii, and” .

(2)  Page 1 Line 19.  “fungal community” to “fungal communities”.

Response: Already modified “fungal community” to “fungal communities”.

(3)  Page 1 Line 19.  “verified the” to “verified that the”.

Response: Already modified “verified the” to “verified that the”.

(4)  Page 1 Line 22.  “fore- and mid-intestine” to “fore- and mid-intestines”.

Response: Already modified “fore- and mid-intestine” to “fore- and mid-intestines”.

(5)  Page 1 Line 22.  “fishes intestines” to “fishes’ intestines”.

Response: Already modified “fishes intestines” to “fishes’ intestines”.

(6)  Page 1 Line 23.  “various intestinal segment” to “various intestinal segments”.

Response: Already modified “various intestinal segment” to “various intestinal segments”.

(7)  Page 1 Line 25.  “microorganisms” to “microorganism”.

Response: Already modified “microorganisms” to “microorganism”.

(8)  Page 1 Line 28.  “further increasing” to “further increased”.

Response: Already modified “further increasing” to “further increased”.

Introduction

(9)  Page 1 Line 33.  “for great” to “for their great”.

Response: Already modified “for great” to “for their great”.

(10)  Page 1 Line 33.  “ecological and” to “ecological, and”.

Response: Already modified “ecological and” to “ecological, and”.

(11)  Page 1 Line 35.  “habitat” to “habitats”.

Response: Already modified “habitat” to “habitats”.

(12)  Page 1 Line 43.  “the complicated interactions” to “the complex interactions”.

Response: Already modified “the complicated interactions” to “the complex interactions”.

(13)  Page 2 Line 46.  For “among the different species”, do you mean “among the different fish species”?

Response: Yes, it means among the different fish species. The “fish” had been added into the words.

(14)  Page 2 Lines 46-47 (and others throughout the manuscript).  “golden pompano” to “Golden Pompano”.  I would suggest that the common names of fish species be capitalized, following the usage in the 7th edition of Common and Scientific Names of Fishes from the United States, Canada, and Mexico (AFS Special Publication 34; 2013).

Response: Already modified “golden pompano” to “Golden Pompano”.

(15)  Page 2 Line 47.  “Spirochaetes and” to “Spirochaetes, and”.

Response: Already modified “Spirochaetes and” to “Spirochaetes, and”.

(16)  Page 2 Line 48.  “barramundi” to “Barramundi”.

Response: Already modified  “barramundi” to “Barramundi”.

(17)  Page 2 Line 50.  “Pseudomonas” to “and Pseudomonas”.

Response: Already modified “Pseudomonas” to “and Pseudomonas”.

(18)  Page 2 Lines 50-51.  “mangrove red snapper” to “Mangrove Red Snapper”.

Response: Already modified “mangrove red snapper” to “Mangrove Red Snapper”.

(19)  Page 2 Line 52.  “Pantoea and” to “Pantoea, and”.

Response: Already modified  “Pantoea and” to “Pantoea, and”.

(20)  Page 2 Line 54.  “Firmicutes and” to “Firmicutes, and”.

Response: Already modified “Firmicutes and” to “Firmicutes, and”.

(21)  Page 2 Line 58.  “able to colonize” to “being able to colonize”.

Response: Already modified “able to colonize” to “being able to colonize”.

(22)  Page 2 Line 60.  “they promote” to “as they promote”.

Response: Already modified “they promote” to “as they promote”.

(23)  Page 2 Line 75.  “fish, however” to “fish. However”.

Response: Already modified “fish, however” to “fish. However”.

(24)  Page 2 Line 77. “in situ” should be italicized.

Response: Thank you for your review,  “in situ” has been modified to italicise

(25)  Page 2 Line 77.  “[24] and” to “[24], and”.

Response: Already modified “[24] and” to “[24], and”.

(26)  Page 2 Line 80.  “golden permits (T. blochii) and” to “Golden Permit (T. blochii), and”.

Response: Already modified “golden permits (T. blochii) and” to “Golden Permit (T. blochii), and”.

(27)  Page 2 Line 81.  “from the South China Sea” to “in the South China Sea”.

Response: Already modified “from the South China Sea” to “in the South China Sea”.

(28)  Page 2 Line 83.  “of three coral reef fishes” to “of the three coral reef fishes”.

Response: Already modified “of three coral reef fishes” to “of the three coral reef fish”. “Fish” is considered more suitable than “fishes” in the manuscript. So many “fishes” have been changed to “fish” in the revised manuscript.

(29)  Page 2 Line 88.  “dissimilarity of the fungal community” to “dissimilarity of the fungal communities”.

Response: Already modified “dissimilarity of the fungal community” to “dissimilarity of the fungal communities”.

(30)  Some additional information such as brief background on the life history, feeding strategies, and potential prey of the three coral reef fish species in the South China Sea, which may be related to their intestinal fungal communities, would be helpful in the Introduction section to set the stage.  I would also suggest emphasizing somewhere in the Introduction section the significance of the comparison of the diversity and antimicrobial activity of culturable fungi from different intestinal segments of multiple species and their surrounding environments in a phylogenetic analysis of ITS sequences framework, because these approaches might help people from broader field find your work useful rather than people just working on the particular species or area being interested in it.

Response: Thank you for your good advice, we have reviewed the relevant literature and have now included the relevant content in the foreword. Which were as follows.

These coral reef fish generally have a complex life history. Spawning and larval settlement occurred in the marine areas of river mouths, and then juvenile fish migrate upstream to fresh water where they grew and became mature as males [28]. They exhibited a relatively high trophic status in the food chain and played a crucial role in maintaining the ecological balance of coral reef systems. They fed on fish and crustaceans, but consumed a variety of algae as well [2].

The information on fungal diversity and distribution in different intestinal segments of coral reef fish is very scarce and important, which provides the baseline data for gut microbiota in coral reef fish from the South China Sea, and a reference for the study on gut microbiota in other marine organisms. In addition, 37 representative fungal isolates were selected for screening their antimicrobial activity against six marine pathogenic microorganisms, which can check the potential of these fungi from different intestinal segments in the production of bioactive molecules against pathogenic microorganisms and provide a good resource for our subsequent screening of marine microbial active substances.

The following references were added into the revised manuscript.

[2] Gao, Y.M.; Zou, K.S.; Zhou, L.; Huang, X.D.; Li, Y.Y.; Gao, X.Y.; Chen, X.; Zhang, X.Y. Deep insights into gut microbiota in four carnivorous coral reef fish from the south china sea. Microorganisms 2020, 8, doi: 10.3390/microorganisms8030426.

[28] McCulloch, M.; Cappo, M.; Aumend, J.; Müller, W. Tracing the life history of individual barramundi using laser ablation mc-icp-ms sr-isotopic and sr/ba ratios in otoliths. Marine and Freshwater Research 2005, 56, 637, doi: 10.1071/MF04184.

Materials and Methods

(31)  Page 3 Line 98.  “an average annual temperature” to “an annual average temperature”.

Response: Already modified “an average annual temperature” to “an annual average temperature”.

(32)  Page 3 Line 99.  “The maximum temperature of the sea surface is” to “The maximum sea surface temperature (SST) is”.  I would suggest using “sea surface temperature”, as SST is a commonly used variable in marine science.

Response: Already modified “The maximum temperature of the sea surface is” to “The maximum sea surface temperature (SST) is”.

(33)  Page 3 Line 102.  “TB3) and” to “TB3), and”.

Response: Already modified “TB3) and” to “TB3), and”.

(34)  Page 3 Lines 104-106.  The intestinal fungal communities of the three coral reef fish species may be affected by the sex, size, and life stages of fish samples collected.  Have you considered that?  I would suggest adding some information on the sex, size, and life stages of fish samples collected in this study.

Response: Thank you very much for your suggestion. What you said is very correct. The intestinal fungal communities of the three coral reef fish species may be affected by the sex, size, and life stages of fish samples collected. We overlooked this detail. We have added relevant data and descriptions in the revised manuscript, which were as follows.

All the coral fishes were mature, and males (with a length of 20-30 cm).

(35)  Page 3 Line 105.  “each species fish” to “each fish species”.

Response: Already modified “each species fish” to “each fish species”.

(36)  Page 3 Line 105.  I would suggest using “nine” rather than “9” when the number is less than ten.

Response: Already modified “nine” to “9”.

(37)  Page 3 Line 106.  “TB3 and” to “TB3, and”.

Response: Already modified “TB3 and” to “TB3, and”.

(38)  Page 4 Line 147.  “agar) and SYA” to “agar), and SYA”.

Response: Already modified “agar) and SYA” to “agar), and SYA”.

(39)  Page 4 Line 151.  “agar and” to “agar, and”.

Response: Already modified “agar and” to “agar, and”.

(40)  Page 4 Line 157.  “taken into in a” to “taken into a”.

Response: Already modified “taken into in a” to “taken into a”.

(41)  Page 4 Line 159.  “watered bath at 65 °C” to “and the mixture was put in a water bath at 65 °C”.

Response: Already modified “watered bath at 65 °C” to “and the mixture was put in a water bath at 65 °C”.

(42)  Page 4 Line 159.  “min, centrifuged at 12000” to “min and then centrifuged at 12,000”.

Response: Already modified “min, centrifuged at 12000” to “min and then centrifuged at 12,000”.

(43)  Page 4 Lines 162-163.  “centrifuged it at 12000 r/min for 5min” to “and centrifuged at 12,000 r/min for 5 min”.

Response: Already modified “centrifuged it at 12000 r/min for 5min” to “and centrifuged at 12,000 r/min for 5 min”.

(44)  Page 4 Line 163.  “an equal volume” to “and an equal volume”.

Response: Already modified “an equal volume” to “and an equal volume”.

(45)  Page 4 Line 164.  “tube, the centrifuge tube” to “tube. The centrifuge tube”.

Response: Already modified “tube, the centrifuge tube” to “tube. The centrifuge tube”.

(46)  Page 4 Line 165.  “12000” to “12,000”.

Response: Already modified “12000” to “12,000”.

(47)  Page 4 Line 189.  “1000” to “1,000”.

Response: Already modified “1000” to “1,000”.

(48)  Page 5 Line 195.  “PP) and” to “PP), and”.

Response: Already modified “PP) and” to “PP), and”.

(49)  Page 5 Line 195.  “[34], and” to “[34]; and”.

Response: Already modified “[34], and” to “[34]; and”.

(50)  Page 5 Line 197.  “AS) and” to “AS), and”.

Response: Already modified “AS) and” to “AS), and”.

(51)  Page 5 Line 198.  “7 days” to “seven days”.

Response: Already modified “7 days” to “seven days”.

(52)  Page 5 Line 207.  “The abundance/species richness” to “The species

Response: Already modified “The abundance/species richness” to “The species

(53)  Page 5 Lines 207-209.  There is little explanation as to why the Bray-Curtis dissimilarity statistic was chosen in this study.  There are some other statistical approaches that have been used in ecology and environmental science when it comes to analyzing species’ relative abundance and making pairwise comparisons.  However, there is little justification as to why the authors only chose this method.  I would suggest adding a brief explanation to justify why this approach and not other also commonly used ones was chosen.  Given the potential of statistical analyses to influence the assessment of species’ relative abundance and compositional comparisons, it would be better to critically assess why some approaches are chosen and their potential and weaknesses for the available data and specific environment considered.

Response: In order to analyze the differences in fungal communities from different intestinal segments of multiple fish species and their surrounding environments, Bray-Curtis dissimilarity was chosen in this study. The Bray-Curtis analysis can show a linear response to the transfer of abundance from a given species in one plot to the same species in another plot in which the species is less abundant. In addition, five other coefficients can show a rather gradual, although nonlinear change along with the transfer of abundances [39]. Recently, this method has been widely applied in the study of the analysis of differences in microbial communities [31, 40]. The species coefficient Bray-Curtis was calculated from the presence (represented by 1)/absence (represented by zero) matrix of the fungi separated from the three fish intestines and the corresponding seawater, using SPSS software for Windows (Version 11.5) [31, 40].

The following references were added into the revised manuscript.

[31] Wu, F.; Chen, B.; Liu, S.; Xia, X.; Gao, L.; Zhang, X.; Pan, Q. Effects of woody forages on biodiversity and bioactivity of aerobic culturable gut bacteria of tilapia (Oreochromis niloticus). Plos One 2020, 15, e235560, doi: 10.1371/journal.pone.0235560.

[39] Ricotta, C.; Podani, J. On some properties of the bray-curtis dissimilarity and their ecological meaning. Ecological Complexity 2017, 31, 201-205, doi: 10.1016/j.ecocom.2017.07.003.

[41] Bass, D.; Howe, A.; Brown, N.; Barton, H.; Demidova, M.; Michelle, H.; Li, L.; Sanders, H.; Watkinson, S.C.; Willcock, S.; Richards, T.A. Yeast forms dominate fungal diversity in the deep oceans. Proceedings of the Royal Society B-Biological Sciences 2007, 274, 3069-3077, doi: 10.1098/rspb.2007.1067.

(54)  Page 5 Lines 207-209.  Did you use raw or transformed values of the species abundance and richness data when applying the statistical analyses?  It would be better to specify a little more even though data transformation may not be necessary for some situations, because it is an important step to make sure the variables meet the underlying assumptions of the algorithms before conducting any statistical analyses.  Statistical approaches with different mechanisms may have different underlying assumptions of normality, linearity or multicollinearity, and some variables may need to be transformed to meet specific assumptions.

Response: The transformed values were used in statistical analysis. For example, if the number of isolates is zero, they are represented by zero, otherwise they are represented by 1. We have already explained in the statistical method, which was as follows. The species coefficient Bray-Curtis was calculated from the presence (represented by 1)/absence (represented by zero) matrix of the fungi separated from the three fish intestines and the corresponding seawater, using SPSS software for Windows (Version 11.5) [31, 40].

The following references were added into the revised manuscript.

[31] Wu, F.; Chen, B.; Liu, S.; Xia, X.; Gao, L.; Zhang, X.; Pan, Q. Effects of woody forages on biodiversity and bioactivity of aerobic culturable gut bacteria of tilapia (Oreochromis niloticus). Plos One 2020, 15, e235560, doi: 10.1371/journal.pone.0235560.

[40] Zhang, X.Y.; Fu, W.; Chen, X.; Yan, M.T.; Huang, X.D.; Bao, J. Phylogenetic analysis and antifouling potentials of culturable fungi in mangrove sediments from techeng isle, china. World Journal of Microbiology & Biotechnology 2018, 34, 90, doi: 10.1007/s11274-018-2470-3.

Results

(55)  Page 7 Line 253.  “community between” to “communities between”.

Response: Already modified “community between” to “communities between”.

(56)  Page 7 Line 254.  “the difference in…was not obvious” is not consistent with the “distinct dissimilarity” in Line 252.

Response: The Bray-Curtis analysis showed distinct dissimilarity of the intestinal fungal communities between the three fishes, which was from 41.2% to 71.4% (Table 2), indicating the difference in intestinal fungal community from the three fishes was obvious.

(57)  Page 7 Line 255.  “from three fish species” to “from the three fish species”.

Response: Already modified “from three fish species” to “from the three fish species”.

(58)  Page 7 Line 259.  “pullulans and” to “pullulans, and”.

Response: Already modified “pullulans and” to “pullulans, and”.

(59)  Page 7 Line 261.  “inundatum and” to “inundatum, and”.

Response: Already modified “inundatum and” to “inundatum, and”.

(60)  Page 7 Line 262.  “jirovecii and” to “jirovecii, and”.

Response: Already modified “jirovecii and” to “jirovecii, and”.

(61)  Page 8 Line 272.  “fungal community” to “fungal communities”.

Response: Already modified “fungal community” to “fungal communities”.

(62)  Page 8 Line 277.  “fungal community” to “fungal communities”.

Response: Already modified “fungal community” to “fungal communities”.

(63)  Page 8 Line 281.  “60.0% and” to “60.0%, and”.

Response: Already modified “60.0% and” to “60.0%, and”.

(64)  Page 8 Line 284.  “argentimaculatus and” to “argentimaculatus, and”.

Response: Already modified “argentimaculatus and” to “argentimaculatus, and”.

(65)  Page 8 Line 285.  “33.3% and” to “33.3%, and”.

Response: Already modified “33.3% and” to “33.3%, and”.

(66)  Page 9 Line 298.  “SCAU247-3 and” to “SCAU247-3, and”.

Response: Already modified “SCAU247-3 and” to “SCAU247-3, and”.

Discussion

(67)  Page 9 Line 310.  “closest relative indicating” to “closest relative, indicating”.

Response: Already modified “closest relative indicating” to “closest relative, indicating”.

(68)  Page 9 Line 313.  “isolates were belonged” to “isolates belonged”.

Response: Already modified “isolates were belonged” to “isolates belonged”.

(69)  Page 9 Line 315.  “and seawater” to “and in seawater”.

Response: Already modified “and seawater” to “and in seawater”.

(70)  Page 9 Line 316.  “tilapia and bighead carp” to “Tilapia and Bighead Carp”.

Response: Already modified “tilapia and bighead carp” to “Tilapia and Bighead Carp”.

(71)  Page 9 Line 319.  “a distinct similarity” is not consistent with the “distinct dissimilarity” in Line 252.

Response: Changed “a distinct similarity” (Page 9 Line 319) to “a distinct dissimilarity” in revised manuscript.

(72)  Page 10 Line 323.  “composition” to “compositions”.

Response: Already modified “composition” to “compositions”.

(73)  Page 10 Line 323.  “factors [41-42]suggested” to “factors. [41-42] suggested”.

Response: Already modified “factors [41-42] suggested” to “factors. [41-42] suggested”.

(74)  Page 10 Line 330.  “fungal community” to “fungal communities”.

Response: Already modified “fungal community” to “fungal communities”.

(75)  Page 10 Line 335.  “dissimilarity between” to “dissimilarity of environmental conditions between”.

Response: Already modified “dissimilarity between” to “dissimilarity of environmental conditions between”.

(76)  Page 10 Line 335.  “them [18]. reported” to “them. [18] reported”.

Response: Already modified “them [18]. reported” to “them. [18] reported”.

(77)  Page 10 Line 340.  “most species” to “most Aspergillus species”.

Response: Already modified “most species” to “most Aspergillus species”. However, another reviewer suggested that the two paragraphs were removed in the revised manuscript.

(78)  Page 10 Line 341.  “materials and” to “materials, and”.

Response: Already modified “materials and” to “materials, and”. However, another reviewer suggested that the two paragraphs were removed in the revised manuscript.

(79)  Page 10 Line 343.  “sediment” to “sediments”.

Response: Already modified “sediment” to “sediments”. However, another reviewer suggested that the two paragraphs were removed in the revised manuscript.

(80)  Page 10 Line 365.  “leads” to “lead”.

Response: Already modified “leads” to “lead”.

(81)  Page 10 Line 366.  “microbial community in the intestine” to “microbial communities in the intestines”.

Response: Already modified “microbial community in the intestine” to “microbial communities in the intestines”.

(82)  Page 10 Lines 366-367.  “honeybees, wild-caught” to “honeybees and wild-caught”.

Response: Already modified “honeybees, wild-caught” to “honeybees and wild-caught”.

(83)  Page 10 Lines 367-368.  “the similarity between community” to “and the similarity between communities”.

Response: Already modified “the similarity between community” to “and the similarity between communities”.

(84)  Page 10 Line 370.  “the fore- and mid-intestine” to “the fore- and mid-intestines”.

Response: Already modified “the fore- and mid-intestine” to “the fore- and mid-intestines”.

(85)  Page 11 Line 376.  “the fore- and mid-intestine” to “the fore- and mid-intestines”.

Response: Already modified “the fore- and mid-intestine” to “the fore- and mid-intestines”.

(86)  Page 11 Line 390.  “and not only causes remarkable” to “and causes not only remarkable”.

Response: Already modified “and not only causes remarkable” to “and causes not only remarkable”.

(87)  Page 11 Line 392.  “5.8% of the percentage of fungi that” to “5.8% of the fungi that”.

Response: Already modified “5.8% of the percentage of fungi that” to “5.8% of the fungi that”.

(88)  Besides surrounding environments, the intestinal fungal communities of the three coral reef fish species in the South China Sea may also be affected by potential factors such as the sex, size, life stages, feeding strategies, and potential prey of these fish species.  I would suggest acknowledging in the Discussion section these additional factors based on literature and their potential influence on the intestinal fungal communities of coral reef fish in the study area.

Response:  the following sentence and cites were added into the Discussion Section.

Besides surrounding environments, the intestinal fungal communities of the three coral reef fish species in the South China Sea may also be affected by potential factors such as the sex, size, life stages, feeding strategies, and potential prey of these fish species. For example, Li et al. revealed that the difference had occurred in the intestinal microbial communities between male and female wild largemouth broze gudgeon [48] . Furthermore, [49] found that the intestinal microflora of Atlantic cod Gadus morhua differed depending on whether the fish were fed fish meal, fermented soy protein or standard soy protein.

The following references were added into the revised manuscript.

[48]   Li, X.; Yan, Q.; Ringø, E.; Wu, X.; He, Y.; Yang, D. The influence of weight and gender on intestinal bacterial community of wild largemouth bronze gudgeon (Coreius guichenoti, 1874). Bmc Microbiology 2016, 16, doi: 10.1186/s12866-016-0809-1.

[49].  Ringø, E.; Sperstad, S.; Myklebust, R.; Refstie, S.; Krogdahl, Å. Characterisation of the microbiota associated with intestine of atlantic cod (Gadus morhua l.): The effect of fish meal, standard soybean meal and a bioprocessed soybean meal. Aquaculture 2006, 261, 829-841, doi: https://doi.org/10.1016/j.aquaculture.2006.06.030.

Conclusions

(89)  Page 11 Line 420.  “abundant diversity” to “high diversity”.

Response: Already modified “abundant diversity” to “high diversity”.

(90)  Page 11 Line 422.  “In a comparison of the” to “In the comparison of the”.

Response: Already modified “In a comparison of the” to “In the comparison of the”.

(91)  Page 11 Line 423.  “of intestinal fungi” to “of fish intestinal fungi”.

Response: Already modified “of intestinal fungi” to “of fish intestinal fungi”.

(92)  Page 11 Line 424.  “the fore and mid-intestine” to “the fore- and mid-intestines”.

Response: Already modified “the fore and mid-intestine” to “the fore- and mid-intestines”.

Tables and Figures

(93)  Page 3 Figure 1 Line 119.  “Lutjanus” to “and Lutjanus”.

Response: Already modified “Lutjanus” to “and Lutjanus”.

(94)  Page 3 Figure 1 Lines 120-121.  Species name should be italicized.

Response: Thank you for your review, Species name has been modified to italicise

(95)  Page 3 Figure 1 Line 122.  “from NCBI and” to “from the National Center for Biotechnology Information (NCBI, Bethesda, MD, USA) and”.

Response: Already modified “from NCBI and” to “from the National Center for Biotechnology Information (NCBI, Bethesda, MD, USA) and”

(96)  Page 3 Figure 1 Line 123.  “JX983354.1” to “and JX983354.1”.

Response: Already modified “JX983354.1” to “and JX983354.1”.

(97)  Page 6 Table 1 Line 233.  “by ITS sequences” to “by internal transcribed spacer (ITS) sequences”.

Response: Already modified “by ITS sequences” to “by internal transcribed spacer (ITS) sequences”.

(98)  Page 6 Table 1 Line 235.  Species name should be italicized.

Response: Thank you for your review, Species name has been modified to italicise.

(99)  Page 7 Figure 2 Line 246.  “the ITS sequences” to “the internal transcribed spacer (ITS) sequences”.

Response: Already modified “the ITS sequences” to “the internal transcribed spacer (ITS) sequences”.

(100)  Page 7 Figure 2 Line 248.  “based on based on” is duplicated.

Response: Thank you for your review, “based on” has been deleted.

(101)  Page 8 Table 2 Line 269.  Species name should be italicized.

Response: Thank you for your review, Species name has been modified to italicise.

(102)  Page 8 Table 3 Line 288.  Species name should be italicized.

Response: Thank you for your review, Species name has been modified to italicise.

(103)  Page 9 Table 4.  I would suggest specifying in the table caption or as a footnote what “PP”, “VA”, “ML”, “AV”, “AS”, and “PC” represent.  A good table or figure caption should make the table or figure understandable without reference to the main text.

Response: I appreciate your suggestion, which I have made clear in the footnotes to the table “PP”, “VA”, “ML”, “AV”, “AS”, and “PC” stand for Pseudoaltermonas piscida, Vibrio alginolyticus, Micrococcus luteus, Aspergillus versicolor, A. sydowii and Penicillium citrinum respectively.

I would suggest minor editing of English language.

Response: Thank you for your good advice. We invited Dr Muhammad Amin (College of Pharmacy, Oregon State University, Corvallis, Or-egon USA) to edit and touch up the revised manuscript in English and the changes have been highlighted in red.

**********************************

All revised contents in text were highlighted in red in the revised manuscript.

Thank you very much for your consideration of our manuscript for potential publication. We look forward to hearing from you soon.

Best Regards.

Sincerely yours,

Dr. Xiaoyong Zhang

University Joint Laboratory of Guangdong Province, Hong Kong and Macao Region on Marine Bioresource Conservation and Exploitation, College of Marine Sciences, South China Agricultural University, Guangzhou 510642, China

E-mail: zhangxiaoyong@scau.edu.cn

Round 2

Reviewer 2 Report

N/A

Author Response

Dear reviewer,

Journal of Fungi,

On behalf of my co-authors, I would like to submit the second revised manuscript entitled “Diversity and antimicrobial activity of intestinal fungi from three species of coral reef fish” (jof-2345361) to Journal of Fungi.

The authors are grateful to the reviewers for their valuable comments and time, and all comments are responded carefully point by point.

**********************************

To Reviewer :

1. Comments and Suggestions for Authors:N/A

2. Response: Thanks to the reviewer for his/her hard work and valuable suggestions, as the good advice has made this article richer and more foreword based.

**********************************

Thank you very much for your consideration of our manuscript for potential publication. We look forward to hearing from you soon.

Best Regards.

Sincerely yours,

Dr. Xiaoyong Zhang

University Joint Laboratory of Guangdong Province, Hong Kong and Macao Region on Marine Bioresource Conservation and Exploitation, College of Marine Sciences, South China Agricultural University, Guangzhou 510642, China

E-mail: zhangxiaoyong@scau.edu.cn

Reviewer 3 Report

The authors have addressed most of the comments.  I just have some minor suggestions below, which I hope could help further improve the clarity of the paper.

For the revised version:

-        (1)  Page 2 Line 88.  “Golden Permits (T. blochii) , ” to “Golden Permit (T. blochii), ”.

-        (2)  Page 3 Lines 140-141.  “by Thunnus thynnus, Thalassoma bifasciatum and” to “by Thunnus thynnus and Thalassoma bifasciatum, and”.

-        (3)  Page 3 Line 142.  “USA) and” to “USA), and”.

-        (4)  Page 4 Line 182.  “it, watered bath at 65 °C for 45 min and then” to “it, and the mixture was put in a water bath at 65 °C for 45 min and then”.

-        (5)  Page 4 Line 188.  “tube, the centrifuge tube was” to “tube. The centrifuge tube was”.

-        (6)  Page 5 Lines 238-239.  “The species coefficient Bray-Curtis was” to “The species Bray-Curtis coefficient was”.

-        (7)  Page 5 Line 240.  “by zero” to “by 0”.

-        (8)  Page 7 Line 271.  “Dissimilarity of fungal community” to “Dissimilarity of fungal communities”.

-        (9)  Page 7 Line 274.  “fungal community between” to “fungal communities between”.

-        (10)  Page 8 Line 286.  “and TB” to “, and TB”.

-        (11)  Page 8 Line 303.  “The dissimilarity of fungal community” to “The dissimilarity of fungal communities”.

-        (12)  Page 9 Line 320.  “and PC” to “, and PC”.

English language is fine.

Author Response

Dear reviewer,

Journal of Fungi,

On behalf of my co-authors, I would like to submit the second revised manuscript entitled “Diversity and antimicrobial activity of intestinal fungi from three species of coral reef fish” (jof-2345361) to Journal of Fungi.

The authors are grateful to the reviewers for their valuable comments and time, and all comments are responded carefully point by point.

**********************************

To Reviewer :

Details of the comments are as follows:

  • Page 2 Line 88.  “Golden Permits ( blochii) , ” to “Golden Permit (T. blochii), ”.

Response: Already modified “Golden Permits (T. blochii) , ” to “Golden Permit (T. blochii), ”.

(2)  Page 3 Lines 140-141.  “by Thunnus thynnusThalassoma bifasciatum and” to “by Thunnus thynnus and Thalassoma bifasciatum, and”.

Response: Already modified “by Thunnus thynnusThalassoma bifasciatum and” to “by Thunnus thynnus and Thalassoma bifasciatum, and”.

(3)  Page 3 Line 142.  “USA) and” to “USA), and”.

Response: Already modified “USA) and” to “USA), and”.

(4)  Page 4 Line 182.  “it, watered bath at 65 °C for 45 min and then” to “it, and the mixture was put in a water bath at 65 °C for 45 min and then”.

Response: Already modified “fore- and mid-intestine” to “it, watered bath at 65 °C for 45 min and then” to “it, and the mixture was put in a water bath at 65 °C for 45 min and then”.

(5)  Page 4 Line 188.  “tube, the centrifuge tube was” to “tube. The centrifuge tube was”.

Response: Already modified “tube, the centrifuge tube was” to “tube. The centrifuge tube was”.

(6) Page 5 Lines 238-239.  “The species coefficient Bray-Curtis was” to “The species Bray-Curtis coefficient was”.

Response: Already modified “The species coefficient Bray-Curtis was” to “The species Bray-Curtis coefficient was”.

(7)  Page 5 Line 240.  “by zero” to “by 0”.

Response: Already modified  “by zero” to “by 0”.

(8)  Page 7 Line 271.  “Dissimilarity of fungal community” to “Dissimilarity of fungal communities”.

Response: Already modified “Dissimilarity of fungal community” to “Dissimilarity of fungal communities”.

(9)  Page 7 Line 274.  “fungal community between” to “fungal communities between”.

Response: Already modified  “fungal community between” to “fungal communities between”.

(10)  Page 8 Line 286.  “and TB” to “, and TB”.

Response: Already modified “and TB” to “, and TB”.

(11)  Page 8 Line 303.  “The dissimilarity of fungal community” to “The dissimilarity of fungal communities”.

Response: Already modified “The dissimilarity of fungal community” to “The dissimilarity of fungal communities”.

(12)  Page 9 Line 320.  “and PC” to “, and PC”.

Response: Already modified “and PC” to “, and PC”.

**********************************

All revised contents in text were highlighted in red in the revised manuscript.

Thank you very much for your consideration of our manuscript for potential publication. We look forward to hearing from you soon.

Best Regards.

Sincerely yours,

Dr. Xiaoyong Zhang

University Joint Laboratory of Guangdong Province, Hong Kong and Macao Region on Marine Bioresource Conservation and Exploitation, College of Marine Sciences, South China Agricultural University, Guangzhou 510642, China

E-mail: zhangxiaoyong@scau.edu.cn